# PROTEUS: Incremental Memory Activation for Long-Context Sequence Modeling

## Abstract

The quadratic cost of attention-based sequence models for long contexts has motivated a growing line of research on *memory-based* models that can compress context into a compact state. However, most existing memory models expose a *static* memory throughout the entire sequence. In practice, this approach is prone to memory being "polluted" by early inputs, and it can saturate and fail to incorporate later context. We study a new paradigm of *incremental memory activation*, where the effective capacity of memory is controlled and progressively expanded as the context grows. By imposing an early bottleneck and introducing fresh capacity over time, this leads to better compression of history and reduces interference. We instantiate this paradigm in PROTEUS, a straightforward mechanism that can be incorporated into a broad class of neural memory architectures at no additional cost. We apply PROTEUS to state-of-the-art models, including Hope-Attention, SWLA, Comba, and Titans, and observe consistent improvements across all of them. Overall, our results show that static memory is suboptimal and that incremental memory activation is a promising direction for long-context management.

## 1. Introduction

The rise of Transformers (Vaswani et al., 2017), pure attention-based architectures, has revolutionized the use cases of machine learning and AI, moving from task-specific models to general-purpose systems (Brown et al., 2020; Kaplan et al., 2020). Softmax attention, as the backbone and critical component of Transformers, mainly responsible for sequence mixing, is a perfect memory module (i.e., a memory that caches every single token in the sequence), in which the memory size linearly scales with sequence length (Ramsauer et al., 2021; Bietti et al., 2024; Behrouz et al., 2025c). While plausible for retrieval tasks, this growing memory comes with a quadratic computational cost with respect to the sequence length, limiting the model's long-context understanding.

To overcome of the quadratic cost of Transformers as well as their limited expressivity in tasks that requires sequential reasoning, modern recurrent neural networks have re-gained popularity in recent years (Katharopoulos et al., 2020; Irie et al., 2021; Sun et al., 2023; 2024; Behrouz et al., 2025e). Contrary to Transformers with growing memory, modern RNNs employs a fixed-size memory (a.k.a. hidden state) to compress the context and so the decoding cost per-token is constant, making their computational cost linear or subquadratic with respect to sequence length. While efficient for longer sequences, the length extrapolation ability of RNNs–an ability to generalize to sequence length different from the one that the model is trained on–has limited their potential for long context understanding (Bai et al., 2023; Kuratov et al., 2024; Hsieh et al., 2024; Tiezzi et al., 2024).

Despite recent efforts on enhancing such capabilities through (meta-learning) memory initialization (Sun et al., 2024; Behrouz et al., 2025e), more robust update rules (Schlag et al., 2021; Von Oswald et al., 2023; Behrouz et al., 2025a), and/or more powerful memory architectures (Sun et al., 2024; Behrouz et al., 2025e; Zhang et al., 2025), still recurrent models face challenges to fully take advantage of their memory capacity, mainly due to the fact that initial tokens has less restrictions to be compressed, while later tokens are required to be highly compressed in order to be stored in the memory. This unbalanced allocation of the memory can make the memory biased toward initial tokens (Barbero et al., 2024; Behrouz et al., 2025b).

In this paper, we study a paradigm of *incremental memory activation* for recurrent neural networks, in which the effective memory capacity is scheduled as a function of context length. Specifically, we argue that imposing a capacity bottleneck early in the sequence and progressively activating additional memory parameters as the context grows leads to better long-context memory management.

[1]Anonymous Institution, Anonymous City, Anonymous Region, Anonymous Country. Correspondence to: Anonymous Author <anon.email@domain.com>.

Preliminary work. Under review by the International Conference on Machine Learning (ICML). Do not distribute.

We instantiate this paradigm with a simple yet effective mechanism, PROTEUS, that can be integrated into modern deep and linear-memory recurrent models. Across a diverse set of benchmarks such as language modeling, common-sense reasoning, long-context understanding, and needle-in-a-haystack tasks, PROTEUS consistently improves performance over strong baselines.

**Contributions.** In summary, our key contributions in this paper are as follows.

- **Incremental memory activation.** We introduce incremental memory activation, a paradigm for controlling *effective* model capacity (e.g., memory or parameters) over the context flow via progressive activation, encouraging early compression while reducing later overwrite and interference.

- **PROTEUS: a general, lightweight mechanism.** We propose PROTEUS, a simple block-wise gating scheme that applies incremental activation to both memory reads and writes, is drop-in for a broad class of associative-memory architectures.

- **Strong results and efficiency.** We show that PROTEUS consistently improves state-of-the-art models (Hope-Attention, SWLA, Comba, Titans) across model scales on a broad range of benchmarks, while being more compute-efficient by updating only activated components.

## 2. Preliminaries

Recent works (Sun et al., 2024; Behrouz et al., 2025e) propose an expressive approach to learning from a context flow (e.g., a sequence) via an *associative memory system* that compresses past history through an optimization procedure, defined as follows.

**Definition 2.1** (Associative Memory). Given a set of keys $\mathcal{K} \subseteq \mathbb{R}^{d_k}$ and values $\mathcal{V} \subseteq \mathbb{R}^{d_v}$, an associative memory is an operator $\mathcal{M}(\cdot)$ that maps the set of keys $\mathcal{K}$ to values in $\mathcal{V}$. To learn such a mapping from data, an objective $\tilde{\mathcal{L}}(\cdot; \cdot)$ measures the quality of the mapping, and $\mathcal{M}$ can be computed by solving:

$$\mathcal{M}^* = \arg\min_{\mathcal{M}} \ \tilde{\mathcal{L}}\big(\mathcal{M}(\mathcal{K}), \mathcal{V}\big). \tag{1}$$

In the sequence modeling setting, $\mathcal{K}$ and $\mathcal{V}$ are constructed from different representation views of the inputs (e.g., key and value projections; see (Sun et al., 2024; Behrouz et al., 2025e)). The memory learns the association between these pairs and functions similarly to a cache with compression. Interestingly, from this perspective, attention is a perfect and lossless memory that stores all keys and values without compression.

This framework is flexible, and recent work explores different choices of the objective function $\tilde{\mathcal{L}}(\cdot; \cdot)$, memory parameterizations (e.g., linear maps or MLPs), and optimization procedures, resulting in a variety of associative-memory architectures (Sun et al., 2024; Behrouz et al., 2025e;a;c). For example, TTT (Sun et al., 2024) uses an $\ell_2$ reconstruction objective with simple gradient descent, while Titans (Behrouz et al., 2025e) highlights the importance of momentum and weight decay in the optimization process.

Despite these variations, existing approaches typically treat the memory capacity $\mathcal{M}$ as *static*: the same set of memory parameters is available throughout the entire context flow. This design choice is rarely questioned, even though controlling a model's effective capacity has long been studied in the literature (Guyon et al., 1991; Hansen & Yu, 2001; Ding et al., 2018; Kaplan et al., 2020). From a compression perspective, bottlenecked information flow can encourage models to retain only task-relevant information, as exemplified by autoencoders (Vincent et al., 2010).

Therefore, learning from context via an associative memory—formulated as an optimization problem—is not exempt from this view. Naïvely exposing the full memory capacity throughout the entire context flow can be suboptimal: early inputs may occupy too many degrees of freedom and "pollute" the memory state, leading to poor compression. Moreover, it can increase interference between what is stored and subsequent context, making it harder to incorporate new information later.

In the following, we first introduce the paradigm of *incremental memory activation* for controlling the effective memory capacity over the context in Section 3. We then instantiate this paradigm in PROTEUS in Section 4, which we incorporate into a broad class of sequence models in Section 5, yielding consistent improvements.

## 3. Incremental Memory Activation

We propose the paradigm of *incremental memory activation*, in which the *effective* memory capacity grows with input position, as opposed to prior work that treats memory capacity as static. This paradigm is motivated by two complementary goals: (i) **early compression**, which forces the model to summarize the earlier context through a restricted-capacity bottleneck; and (ii) **fresh capacity**, which unlocks previously unused memory components later in the context to reduce overwrite and interference while continuing to leverage what has already been learned.

We now modify the associative memory definition in 2.1 to incorporate an activation operator $\mathcal{G}_t$ that controls the effective memory capacity at time step $t$.

**Definition 3.1** (Capacity-Scheduled Associative Memory). Given keys $\mathcal{K}_{\leq t}$ and values $\mathcal{V}_{\leq t}$ observed up to position

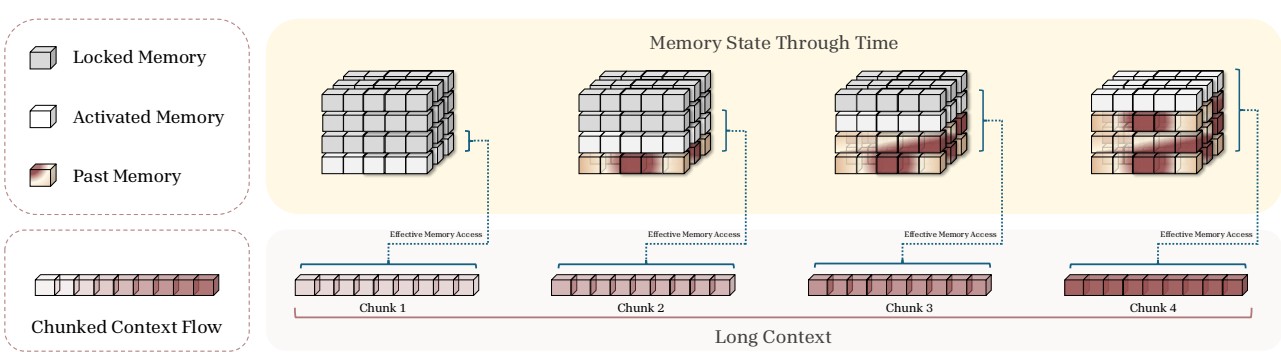

*Figure 1.* **PROTEUS** incrementally expands the *active* subset of the memory state as the context progresses. At each step, the model reads from and writes to the currently activated blocks together with the previously activated blocks, while the remaining (not-yet-activated) blocks stay locked. As time advances, additional blocks are unlocked, providing fresh capacity for later tokens and reducing overwrite and interference.

$t$, a capacity-scheduled associative memory is an operator $\mathcal{G}_t(\mathcal{M}(\cdot))$ that maps $\mathcal{K}_{\leq t}$ to $\mathcal{V}_{\leq t}$ subject to constraints imposed on the memory by the activation operator $\mathcal{G}_t$. To learn such a mapping from data, an objective $\tilde{\mathcal{L}}(\cdot;\cdot)$ measures its quality, and $\mathcal{M}_t(\cdot)$ can be computed by solving:

$$\mathcal{M}_t^* = \arg\min_{\mathcal{M}} \tilde{\mathcal{L}}\Big(\mathcal{G}_t(\mathcal{M})(\mathcal{K}_{\leq t}),\ \mathcal{V}_{\leq t}\Big)$$
$$\text{s.t.} \quad \mathcal{M}_t^{(g)} = \mathcal{G}_t(\mathcal{M}). \tag{2}$$

Here, $\mathcal{M}$ denotes the memory with total capacity, and $\mathcal{M}_t^{(g)}$ denotes the subset of its components selected by the activation operator at time step $t$. Setting $\mathcal{G}_t$ to the identity recovers the standard associative memory definition, and this formulation is general.

This modification supports the two goals above. With an appropriate activation operator, it always limits effective capacity throughout the context and thereby encourages compression through an explicit bottleneck. As additional capacity is unlocked over time, it introduces fresh degrees of freedom, which reduces interference between previously stored information and new updates.

Our incremental activation can be instantiated in multiple ways. In particular, it requires specifying an activation operator $\mathcal{G}_t$ that maps the position $t$ to an effective subset of the memory. In this work, we study a simple instantiation and leave richer, data-dependent activation policies to future work.

## 4. PROTEUS

We now introduce PROTEUS, our concrete realization of incremental memory activation. We first present it for *memory-based* recurrent sequence models, where an explicit associative memory state is updated online over the sequence. We then broaden the scope using the recent Nested Learning (NL) perspective (Behrouz et al., 2025d), which views

pretraining gradient-based optimization itself as an instance of associative memory. Under this lens, we can apply the same incremental activation principle to all model parameters—not just memory. This allows PROTEUS-style scheduling to extend beyond recurrent memory models and to be instantiated in architectures such as Hope-Attention.

### 4.1. PROTEUS for Memory

PROTEUS follows a simple block-wise design. We **partition** the memory parameters into $E$ equal-sized contiguous blocks. We then define a deterministic **activation schedule** that maps each input position $t$ to a set of active blocks, increasing stepwise over the context until all $E$ blocks are enabled (see Figure 1). Operationally, PROTEUS realizes this schedule via a lightweight masking operator that gates both memory *reads* and *writes*, so that inactive blocks remain locked while activated blocks participate in retrieval and updates.

We now review the generic *online* memory update and retrieval procedure, and then define the gating operator to obtain gated versions of the update and retrieval steps.

**Generic online memory update and retrieval.** Following prior work (Sun et al., 2024; Behrouz et al., 2025e), we represent the neural memory as a state $\mathcal{M}_t$ that is updated online as each new input $x_t$ (e.g., a token) arrives. A common instantiation uses a gradient-based update,

$$\mathcal{M}_t = \mathcal{M}_{t-1} - \theta_t \nabla\ell(\mathcal{M}_{t-1}; x_t), \tag{3}$$

where $\ell$ is typically a self-supervised loss and $\theta_t$ is a step size. In this view, the gradient $\nabla\ell(\mathcal{M}_{t-1}; x_t)$ provides a "surprise" signal indicating how strongly the memory should be modified in response to $x_t$. More generally, we can write the memory update abstractly as $\mathcal{M}_t = \text{Upd}(\mathcal{M}_{t-1}; x_t)$, where $\text{Upd}(\cdot)$ may implement a delta rule, a momentum/weight-decay update, or an inner-loop opti-

mizer; however, for simplicity, we stick to standard gradient descent here.

At each step $t$, the model reads from the current memory state to produce a memory-conditioned output (or representation),

$$y_t \;=\; \mathrm{Read}(\mathcal{M}_{t-1}, x_t), \tag{4}$$

where $\mathrm{Read}(\cdot)$ depends on the memory architecture (e.g., a forward path through an MLP).

**Gating operator.** Most modern neural memory architectures parameterize memory as a linear map or MLPs, and thus the activation operator can be implemented as simple multiplicative gating of the memory parameters. Concretely, we lift the block activation into an elementwise mask $g_t \in \{0,1\}^{\dim(\mathcal{M})}$ that enables exactly the parameters belonging to the active blocks, and define the gated memory as

$$\mathcal{M}_t^{(g)} \;:=\; \mathcal{G}_t(\mathcal{M}_t) \;:=\; g_t \odot \mathcal{M}_t, \tag{5}$$

where $\odot$ denotes elementwise multiplication. Intuitively, $\mathcal{M}_t^{(g)}$ is the *effective* memory exposed at position $t$: inactive components are masked out (locked) and do not participate in retrieval or learning.

**Gated online memory update and retrieval.** Using the gradient-based update in Equation (3), PROTEUS applies the update only to active components while leaving inactive blocks unchanged:

$$\mathcal{M}_t = \mathcal{M}_{t-1} - \theta_t\big(g_t \odot \nabla\ell(\mathcal{M}_{t-1}^{(g)}; x_t)\big), \tag{6}$$

where $\mathcal{M}_{t-1}^{(g)} = \mathcal{G}_t(\mathcal{M}_{t-1})$.

Equivalently, one can view PROTEUS as updating the gated state and then embedding it back into the full memory with locked components preserved:

$$\mathcal{M}_t = \mathcal{M}_{t-1} + \mathcal{G}_t\big(\Delta\mathcal{M}_t\big), \tag{7}$$

where $\Delta\mathcal{M}_t := -\theta_t\nabla\ell(\mathcal{M}_{t-1}^{(g)}; x_t)$.

Retrieval is gated in the same way: the model reads only from the active subspace,

$$y_t \;=\; \mathrm{Read}\big(\mathcal{M}_{t-1}^{(g)}, x_t\big) \;=\; \mathrm{Read}\big(\mathcal{G}_t(\mathcal{M}_{t-1}), x_t\big). \tag{8}$$

Thus, PROTEUS enforces that both *writes* (memory updates) and *reads* (memory access) are limited to the currently active blocks, while previously locked blocks remain untouched until they are activated.

Finally, we need to specify a mapping from an input position $t$ to a set of active memory blocks.

**Memory expansion schedule.** We adopt a simple deterministic schedule that uniformly increases the number of active blocks as a function of position in the context. Let $N$ denote the maximum context length and let $E$ be the number of equal-sized blocks in the memory partition. For notational simplicity, we assume $\dim(\mathcal{M})$ is divisible by $E$; otherwise, the last block can absorb the remainder. Define the block size $d := \dim(\mathcal{M})/E$ and a (uniform) step length

$$\Delta \;:=\; \max\Big(1,\; \big\lfloor \tfrac{N}{E} \big\rfloor\Big). \tag{9}$$

At time step $t \in \{1, \ldots, N\}$, we activate the first

$$k(t) \;:=\; \min\left(E,\; 1 + \left\lfloor \frac{t-1}{\Delta} \right\rfloor\right) \tag{10}$$

blocks, so the active fraction of memory increases in $E$ uniform increments across the context. The corresponding mask $g_t \in \{0,1\}^{\dim(\mathcal{M})}$ is a prefix mask that enables exactly the components belonging to the first $k(t)$ blocks:

$$g_t[j] \;=\; \begin{cases} 1, & 1 \le j \le k(t)\,d, \\ 0, & \text{otherwise.} \end{cases} \tag{11}$$

Thus, tokens in the first segment $t \in [1, \Delta]$ update and read only the first $1/E$ fraction of the memory; tokens in the next segment $t \in [\Delta+1, 2\Delta]$ use the first $2/E$ fraction; and so on, until the full memory becomes available after the final expansion step. See Figure 1 for an illustration.

### 4.2. PROTEUS for Pre-training

In the previous section, we mainly focused on the use of PROTEUS for enhancing the memory management of associative memory module. In fact, given the context of the model, PROTEUS schedules the active parameters in the fixed-size memory, allowing the memory to have fresh capacity for the later tokens in the context. In this section, we use Nested Learning (NL) perspective (Behrouz et al., 2025d) to adapt the idea of PROTEUS for pre-training. More specifically, NL shows that pre-training is a form of parametric in-context learning (i.e., backpropagation being a form of associative memory), where the model aims to compress the pre-training data into its parameters. Therefore, throughout the pre-training phase, the model learns less from initial tokens, mainly due to the fact that it is overparameterized and so does not need to compress the data. To overcome this issue, we adapt PROTEUS for the pre-training, where we schedule the MLP blocks' parameters.

Given an architecture $A$, with MLP blocks of $\{\mathrm{MLP}_i\}_{i=1}^{L'}$ parameterized with $\{\boldsymbol{\theta}_i\}_{i=1}^{L'}$, and a scheduling mask $\{g_t\}_{i=1}^{|D|}$ (mask $g_t$ determines the active parameters of the model for $t$-th data sample in the dataset.), we update the model with

$$\boldsymbol{\theta}_{i+1} = \boldsymbol{\theta}_i - \eta_{i+1}\,(g_{i+1} \odot \boldsymbol{e}_i), \tag{12}$$

where $e_i$ indicates the error, computed by the optimizer. As an example, when using AdamW optimizer, PROTEUS for pre-training updates the weights as:

$$\boldsymbol{\theta}_{i+1} = \alpha_{i+1}\,\boldsymbol{\theta}_i - \eta_{i+1}\,\frac{\hat{\boldsymbol{m}}_{i+1}}{\sqrt{\hat{\boldsymbol{v}}_{i+1}} + \varepsilon}, \qquad (13)$$

where $\hat{\boldsymbol{m}}_{i+1} = \frac{\boldsymbol{m}_{i+1}}{1-\beta_1}$ and $\hat{\boldsymbol{v}}_{i+1} = \frac{\boldsymbol{v}_{i+1}}{1-\beta_2}$ with:

$$\boldsymbol{m}_{i+1} = \beta_1\,\boldsymbol{m}_i + (1-\beta_1)\,\left(g_{i+1} \odot \nabla_{\boldsymbol{\theta}_i} L(\boldsymbol{\theta}_i, \boldsymbol{x}_{i+1})\right),$$

$$\boldsymbol{v}_{i+1} = \beta_2\,\boldsymbol{m}_i + (1-\beta_2)\,\left(g_{i+1} \odot \left(\nabla_{\boldsymbol{\theta}_i} L(\boldsymbol{\theta}_i, \boldsymbol{x}_{i+1})\right)^2\right),$$

Given this formulation, only the weights that are not masked by $g_{i+1}$ are updated by the AdamW. With scheduling the mask matrices $\{g_i\}_{i=1}^{|D|}$, we can control the parameters that are active for each data sample. In this formulation, we make sure that there are fresh capacity for each set of the data batches, and also do not overparameterize the model at its initial phase of training.

**Initialization and the Importance of PROTEUS for Continual Learning.** To unlock the continual learning capabilities of the LLMs, there are two important steps: (1) The model needs to adapt fast to the context and update itself accordingly, and (2) These new updates must not cause forgetting about the previous capabilities, mainly referred to as catastrophic forgetting. Starting from an initial state for MLP blocks (e.g., initializing from a pre-trained Transformers (Behrouz et al., 2025d)) ensures that the model has good general capabilities but to satisfy (1), it needs to be updated in-context. Although simply updating all parameters can cause the model to diverge from its initial state, causing catastrophic forgetting, PROTEUS can avoid this by only allowing the parameter update for a small part of the model's parameters.

**Hope-Attention.** As a proof of concept, here we use PROTEUS on top of Hope architecture (Behrouz et al., 2025d). The design of Hope architecture with softmax attention is very similar to Transformers, where instead of one MLP blocks after each attention module, Hope uses multiple MLP blocks, each of them are updated with different frequencies. More formally, given the frequency of update for each block $f_i$ and input $\boldsymbol{x}$, the output of Hope with softmax attention is computed as:

$$\boldsymbol{k}_t = \boldsymbol{x}_t W_{\boldsymbol{k}}, \qquad \boldsymbol{v}_t = \boldsymbol{x}_t W_{\boldsymbol{v}}, \qquad \boldsymbol{q}_t = \boldsymbol{x}_t W_{\boldsymbol{q}} \quad (14)$$

$$\boldsymbol{h}_t = \texttt{Attn}\left(\{\boldsymbol{k}_i\}_{i=1}^t,, \{\boldsymbol{v}_i\}_{i=1}^t, \boldsymbol{q}_t\right) \quad (15)$$

$$\boldsymbol{y}_t = \texttt{MLP}^{(f_k)}(\texttt{MLP}^{(f_{k-1})}(\cdots\texttt{MLP}^{(f_1)}(\boldsymbol{h}_t))), \quad (16)$$

where the parameters of $\ell$-th MLP block, i.e., $\boldsymbol{\theta}^{(f_\ell)}$, are updated every $C^{(\ell)}$ steps:

$$\boldsymbol{\theta}_{i+1}^{(f_\ell)} = \boldsymbol{\theta}_i^{(f_\ell)} - \boldsymbol{e}_i, \quad (17)$$

where

$$\boldsymbol{e}_i = \begin{cases} \sum_{t=i-C^{(\ell)}}^i \eta_t^{(\ell)} f(\boldsymbol{\theta}_t^{(f_\ell)}; \boldsymbol{x}_t) & \text{if } i \equiv 0 \ (\texttt{mod } C^{(\ell)}), \\ 0 & \text{otherwise.} \end{cases}$$

In this formulation, $f(.)$ is the error of the optimization process, which depends on the choice of optimizer and also the objective. Often for language modeling, this choice is the pair of Next Token Prediction (NTP) as the objective and AdamW as the optimizer.

# 5. Experiments

**Experimental Setup.** We apply our framework to four families of high-performing models: Hope-Attention (Behrouz et al., 2025d), Sliding-Window Linear Attention (SWLA) (Behrouz et al., 2025a), Comba (Hu et al., 2025), and Titans (Behrouz et al., 2025e). For simplicity, we set the number of partition blocks in PROTEUS to $E = 4$. For reference, we also report results for Transformer++ (Touvron et al., 2023), RetNet (Sun et al., 2023), and DeltaNet (Yang et al., 2024c). All models are trained on FineWeb (Penedo et al., 2024) with an 8K training context window. We consider two parameter scales (760M and 1.3B), training on 50B and 100B tokens, respectively, and report perplexity on held-out validation data. We use AdamW with a learning rate of $4 \times 10^{-4}$ and a cosine annealing schedule, a batch size of 0.5M tokens, and weight decay of 0.1. For downstream evaluation, we benchmark the trained models on Wikitext (Merity et al., 2017), LAMBADA (LMB) (Paperno et al., 2016), PIQA (Bisk et al., 2020), HellaSwag (Zellers et al., 2019), WinoGrande (Sakaguchi et al., 2021), ARC-Easy (ARC-e) and ARC-Challenge (ARC-c) (Clark et al., 2018), SIQA (Sap et al., 2019), and BoolQ (Clark et al., 2019).

## 5.1. Language Modeling and Common-Sense Reasoning

Results for PROTEUS variants of four strong architectures, together with additional baselines, are summarized in Table 1 at two scales. Across all four model families and at both scales, adding PROTEUS consistently improves performance over the corresponding base model, yielding lower perplexity and higher average downstream accuracy.

At 760M, PROTEUS improves language modeling perplexity for each architecture and also increases the average common-sense score (e.g., Hope-Attention: Avg. $53.15 \to 53.66$; Comba: $51.43 \to 51.98$; Titans: $52.65 \to 53.09$). Moreover, the best overall 760M results in Table 1 are achieved by Hope-Attention+PROTEUS, attaining the lowest perplexities (Wiki. 20.23, LMB. 20.90) and the highest average accuracy (53.66). At 1.3B, the gains persist and become particularly pronounced for the strongest memory model: Titans+PROTEUS achieves the best overall performance in

*Table 1.* **Language modeling and commonsense reasoning results.** We report perplexity on Wikitext and LAMBADA and accuracy on seven commonsense benchmarks; Avg. denotes the mean accuracy across reasoning tasks. Results are shown for 760M (50B tokens) and 1.3B (100B tokens) models, comparing each backbone with and without PROTEUS (shaded rows).

| Model | Wiki. ppl ↓ | LMB. ppl ↓ | LMB. acc ↑ | PIQA acc ↑ | Hella. acc_n ↑ | Wino. acc ↑ | ARC-e acc ↑ | ARC-c acc_n ↑ | SIQA acc ↑ | BoolQ acc ↑ | Avg. ↑ |
|---|---|---|---|---|---|---|---|---|---|---|---|
| | | | | | 760M params / 50B tokens | | | | | | |
| Transformer++ | 22.08 | 22.41 | 38.2 | 68.9 | 44.1 | 56.8 | 67.0 | 34.9 | 40.2 | 62.4 | 51.56 |
| Hope-Attention | 20.67 | 21.36 | 39.5 | 70.4 | 50.1 | 56.3 | 66.9 | 37.5 | 40.7 | 63.8 | 53.15 |
| +PROTEUS | 20.23 | 20.90 | 40.2 | 70.3 | 51.3 | 57.0 | 67.2 | 38.3 | 41.4 | 63.6 | **53.66** |
| RetNet | 23.54 | 23.87 | 35.7 | 66.4 | 42.8 | 53.6 | 64.8 | 33.1 | 38.7 | 57.4 | 49.06 |
| SWLA ($c = 2$) | 22.76 | 22.85 | 36.7 | 67.5 | 44.2 | 54.7 | 64.6 | 34.3 | 39.9 | 59.1 | 50.12 |
| +PROTEUS | 22.19 | 21.32 | 37.0 | 68.6 | 44.3 | 54.5 | 66.3 | 34.5 | 40.4 | 59.6 | **50.65** |
| DeltaNet | 22.89 | 23.13 | 37.2 | 68.0 | 45.2 | 52.9 | 65.4 | 33.4 | 40.2 | 59.8 | 50.26 |
| Comba | 21.94 | 21.77 | 38.4 | 67.1 | 47.1 | 53.0 | 65.9 | 35.5 | 40.8 | 63.7 | 51.43 |
| +PROTEUS | 21.38 | 21.45 | 39.0 | 67.9 | 47.8 | 54.2 | 66.3 | 35.2 | 41.3 | 64.1 | **51.98** |
| Titans | 20.92 | 21.28 | 39.3 | 69.2 | 49.9 | 53.1 | 67.3 | 36.9 | 41.7 | 63.8 | 52.65 |
| +PROTEUS | 20.76 | 21.01 | 39.6 | 69.8 | 50.4 | 54.3 | 67.5 | 36.7 | 42.6 | 63.8 | **53.09** |
| | | | | | 1.3B params / 100B tokens | | | | | | |
| Transformer++ | 17.04 | 17.42 | 45.3 | 73.1 | 51.8 | 59.3 | 71.0 | 38.2 | 43.5 | 64.3 | 55.81 |
| Hope-Attention | 15.66 | 13.39 | 48.2 | 72.9 | 55.0 | 59.8 | 72.2 | 38.7 | 43.3 | 65.4 | 56.93 |
| +PROTEUS | 15.78 | 13.24 | 48.4 | 73.1 | 55.2 | 60.7 | 72.8 | 38.5 | 44.1 | 64.9 | **57.21** |
| RetNet | 18.84 | 17.29 | 40.8 | 70.6 | 48.9 | 56.0 | 67.9 | 35.3 | 41.6 | 62.7 | 52.98 |
| SWLA ($c = 2$) | 18.11 | 16.95 | 40.7 | 71.4 | 49.2 | 57.3 | 68.8 | 37.0 | 42.4 | 62.6 | 53.67 |
| +PROTEUS | 18.19 | 17.10 | 40.2 | 71.8 | 50.5 | 57.6 | 69.5 | 36.7 | 43.0 | 63.1 | **54.05** |
| DeltaNet | 17.39 | 17.02 | 40.0 | 71.4 | 50.1 | 53.9 | 68.3 | 36.6 | 43.3 | 60.7 | 53.04 |
| Comba | 16.98 | 14.17 | 43.8 | 73.2 | 53.5 | 60.1 | 70.9 | 39.2 | 44.2 | 58.5 | 55.42 |
| +PROTEUS | 17.03 | 14.09 | 44.5 | 73.1 | 53.9 | 60.4 | 71.2 | 39.2 | 44.6 | 60.1 | **55.88** |
| Titans | 15.36 | 13.18 | 50.9 | 74.0 | 54.6 | 57.3 | 72.2 | 40.9 | 42.6 | 63.1 | 56.95 |
| +PROTEUS | 15.07 | 12.99 | 51.5 | 75.8 | 54.4 | 58.8 | 72.3 | 41.4 | 43.8 | 63.9 | **57.74** |

the table, with the lowest perplexities (Wiki. 15.07, LMB. 12.99) and the highest average accuracy (57.74). These improvements hold even when compared against competitive non-memory baselines such as Transformer++, indicating that the benefit is not tied to a single architecture but rather to how memory is managed throughout the context.

We attribute these gains to PROTEUS 's incremental activation of memory capacity (Sec. 4), which yields more reliable compression in short contexts and better retention in long contexts.

### 5.2. Needle-in-Haystack

We evaluate long-context retrieval on Needle-in-a-Haystack (NIAH); results are in Table 2. Across single-needle and multi-needle settings, PROTEUS improves accuracy over the corresponding base model, with gains that are generally larger at longer contexts and on harder variants. For instance, at 16K PROTEUS improves Titans on S-NIAH-3 (21.4 → 29.8) and S-NIAH-2 (69.4 → 74.2), and im-

proves Comba on S-NIAH-2 (13.4 → 21.2). Similarly, on multi-needle benchmarks at 16K, PROTEUS improves MK-NIAH-1 (Titans 11.8 → 16.8), MQ-NIAH (Hope-Attention 30.6 → 34.2), and MV-NIAH (Hope-Attention 23.0 → 25.8).

### 5.3. Long Context Understanding

In this section, we evaluate the effectiveness of PROTEUS on the long-context understanding of the base model. To this end, we use two common types of tasks: (1) retrieval, and (2) LongBench benchmark tasks:

**Retrieval Tasks.** Following recent studies (Hu et al., 2025; Yang et al., 2024a; Arora et al., 2024; Behrouz et al., 2025a), we evaluate the effect of PROTEUS on the Hope-Attention, Comba, and Titans baselines on recall-intensive (retrieval) tasks from Lockard et al. (2019); Rajpurkar et al. (2016); Arora et al. (2024). The results are reported in Figure 2. Across all datasets and baselines, PROTEUS is more robust at longer context lengths and generally improves length ex-

*Table 2.* Needle-In-A-Haystack experiments with: (1) Single needle with three levels of difficulty: single-needle tasks—S-NIAH-1 (passkey retrieval), S-NIAH-2 (numerical needle), and S-NIAH-3 (UUID-based needle); (2) multi-query; (3) multi-key; and (4) multi-value settings of the benchmark.

| | S-NIAH-1 (pass-key retrieval) | | | S-NIAH-2 (number in haystack) | | | S-NIAH-3 (uuid in haystack) | | |
|---|---|---|---|---|---|---|---|---|---|
| **Model** | 4K | 8K | 16K | 4K | 8K | 16K | 4K | 8K | 16K |
| Transformer | 97.8 | 86.2 | 83.8 | 100 | 99.2 | 96.4 | 82.2 | 72.0 | 48.4 |
| Hope-Attention | 100 | 100 | 100 | 100 | 99.4 | 96.8 | 83.4 | 72.8 | 50.0 |
| +PROTEUS | 100 | 100 | 100 | 100 | 99.4 | 97.2 | 83.2 | 73.2 | 54.4 |
| Comba | 100 | 100 | 99.4 | 92.6 | 47.2 | 13.4 | 62.4 | 13.8 | 7.4 |
| +PROTEUS | 100 | 100 | 99.2 | 91.8 | 49.4 | 21.2 | 60.8 | 18.2 | 10.8 |
| Titans | 100 | 100 | 100 | 99.6 | 84.6 | 69.4 | 74.2 | 42.8 | 21.4 |
| +PROTEUS | 100 | 100 | 100 | 99.2 | 85.2 | 74.2 | 74.2 | 44.0 | 29.8 |

| | MK-NIAH-1 (multi-key line retrieval) | | | MQ-NIAH (multi-query) | | | MV-NIAH (multi-value) | | |
|---|---|---|---|---|---|---|---|---|---|
| **Model** | 4K | 8K | 16K | 4K | 8K | 16K | 4K | 8K | 16K |
| Transformer | 80.0 | 79.4 | 60.8 | 57.4 | 46.2 | 28.8 | 36.6 | 33.8 | 20.2 |
| Hope-Attention | 81.4 | 85.2 | 62.2 | 62.0 | 47.6 | 30.6 | 37.2 | 36.2 | 23.0 |
| +PROTEUS | 80.8 | 85.2 | 64.8 | 62.2 | 48.4 | 34.2 | 36.6 | 36.0 | 25.8 |
| Comba | 22.8 | 20.4 | 10.4 | 22.2 | 16.0 | 6.2 | 15.8 | 14.2 | 6.6 |
| +PROTEUS | 23.4 | 20.2 | 14.2 | 23.8 | 16.6 | 8.4 | 14.8 | 14.0 | 8.2 |
| Titans | 27.0 | 22.8 | 11.8 | 23.2 | 18.4 | 10.2 | 24.8 | 15.4 | 8.8 |
| +PROTEUS | 28.4 | 22.6 | 16.8 | 23.2 | 18.6 | 12.0 | 24.6 | 14.8 | 10.2 |

trapolation. Notably, the performance gain over the baseline increases with context length.

**LongBench Benchmark.** Next, we evaluate the effect of PROTEUS in LongBench benchmark (Bai et al., 2023). The results are reported in Table 3. PROTEUS enhances the performance of all baselines in most cases, and outperforms the base model on average for all the three types of baselines we have selected. Notably, such performance improvement is achieved without any computational overhead.

Overall, these results support our thesis that treating memory capacity as static is suboptimal, and that incremental activation provides a simple and effective mechanism for better long-context management.

# 6. Related Work

**Modern Linear Recurrent Neural Networks.** To address the quadratic complexity of Transformers, recent work has focused on efficient recurrent architectures that enable rapid training and inference (Tiezzi et al., 2024; Katharopoulos et al., 2020; Yang et al., 2024b;c). Early linear recurrent models, including linear attention (Katharopoulos et al., 2020), RetNet (Sun et al., 2023), RWKV (Peng et al., 2023), and S5 (Smith et al., 2023), utilized data-independent tran-

sition matrices with Hebbian update mechanisms. These evolved into second-generation architectures incorporating input-dependent parameters (Hasani et al., 2023; Peng et al., 2024; Yang et al., 2024a;b) and more expressive memory rules based on the delta rule (Schlag et al., 2021; Yang et al., 2024c;a; Liu et al., 2024; Peng et al., 2025). Recent iterations have scaled these mechanisms to deeper models using momentum-based update rule (Behrouz et al., 2025e) or delta-rule-like updates with non-linearity (Sun et al., 2024). Notably, Siems et al. (2025) recently proposed applying multiple gradient descent steps per token to enhance state tracking. Parallel to these linear approaches, significant progress has been made in optimizing the training and performance of RNNs with non-linear recurrence (Behrouz et al., 2025c; Csordás et al., 2024; Merrill et al., 2024; Lim et al., 2024; Schöne et al., 2025; Karami & Mirrokni, 2025; Von Oswald et al., 2023; Gonzalez et al., 2024; Behrouz et al., 2025d; Irie et al., 2022). Concurrent works such as log-linear attention, memory caching, and RAT have focused on designing recurrent models that their effective memory state grows with sequence length (Anonymous, 2026; Guo et al., 2025; Wei et al., 2025).

**Fast Weight Programs.** The paradigm of treating linear layers as dynamic, writable memory traces its origins to Hopfield networks (Hopfield, 1982). This concept was for-

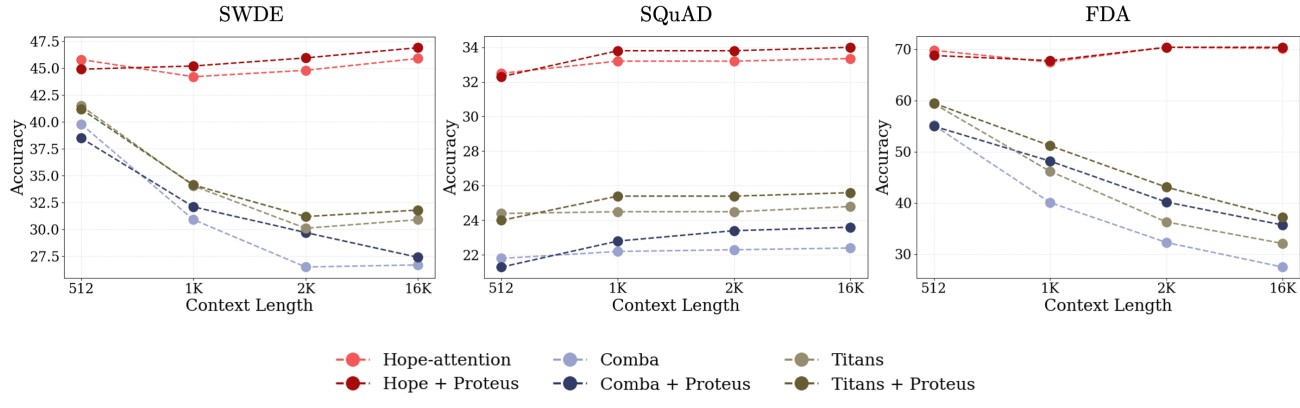

*Figure 2.* **Long-context retrieval.** Accuracy vs. context length on SWDE(Lockard et al., 2019), SQuAD (Rajpurkar et al., 2016), and FDA (Arora et al., 2024) for Hope-Attention, Comba, and Titans. PROTEUS consistently improves robustness at longer contexts, mitigating degradation (notably for Comba and Titans) and improving Hope-Attention across lengths.

*Table 3.* **LongBench evaluation.** Performance on LongBench (Bai et al., 2023) across six long-context tasks (Narrative, Qasper, MultiField, Hotpot, 2WikiMulti, Musique). PROTEUS improves the average score for all three base models, indicating better long-context understanding.

| Model | Narrative | Qasper | MultiField | Hotpot | 2WikiMulti | Musique | Average |
|---|---|---|---|---|---|---|---|
| Hope-Attention | 12.1 | 9.4 | 19.2 | 20.6 | 26.7 | 6.3 | 15.72 |
| + PROTEUS | 12.8 | 10.2 | 20.4 | 21.3 | 27.2 | 7.5 | **16.57** |
| Comba | 7.6 | 10.4 | 15.9 | 14.6 | 23.1 | 6.7 | 13.05 |
| + PROTEUS | 7.6 | 10.7 | 16.1 | 14.5 | 23.3 | 6.8 | **13.17** |
| Titans | 8.0 | 10.3 | 17.4 | 15.6 | 24.8 | 6.7 | 13.8 |
| + PROTEUS | 8.5 | 10.2 | 17.9 | 16.3 | 25.2 | 6.7 | **14.13** |

malized in fast weight programmers (Schmidhuber, 1992; 1993), where a slow network generates weights for a fast network to process sequences. Two primary learning rules dominate this landscape: Hebbian learning (Hebb, 2005) and the delta rule (Prados & Kak, 1989). These mechanisms allow the network to act as a key-value associative memory and have been extensively studied as efficient alternatives to standard attention (Munkhdalai & Yu, 2017; Munkhdalai et al., 2019; Irie et al., 2021; Schlag et al., 2021; Yang et al., 2024c;a).

**Hopfield Networks.** We ground our architecture in the principles of associative memory, originally framed by Hopfield (1982) as energy minimization for storing key-value pairs. While classical Hopfield networks were limited by low storage capacity, modern continuous variants have addressed these bottlenecks through dense associative memories and exponential energy kernels (Krotov & Hopfield, 2016; Krotov, 2021; Li et al., 2024; Lucibello & Mézard, 2024). These "modern Hopfield networks" have gained renewed prominence due to their theoretical equivalence to the attention mechanism in Transformers, a relationship established in recent literature (Ramsauer et al., 2021; Hu et al., 2024).

## 7. Conclusion

We introduced *incremental memory activation*, a simple paradigm for long-context sequence modeling in which the *effective* memory capacity is progressively expanded over the context rather than exposed statically from the start. We instantiated this idea in PROTEUS, a lightweight, blockwise gating mechanism that restricts both memory reads and writes to an active subset and unlocks fresh capacity over time. Across a diverse set of architectures, including Hope-Attention, SWLA, Comba, and Titans, and evaluations spanning language modeling, commonsense reasoning, needle-in-a-haystack, retrieval, and LongBench, PROTEUS consistently improves long-context robustness and length extrapolation, with the largest gains appearing at long context lengths. These results suggest that treating memory capacity as static is suboptimal, and that scheduling effective capacity is a practical and broadly applicable tool for managing long-context computation. Promising directions include learning data-dependent activation policies, combining activation schedules with adaptive compute and retrieval, and extending incremental activation to other parameter subsets and continual adaptation settings.

## Impact Statement

This paper presents work whose goal is to advance the field of machine learning. There are many potential societal consequences of our work, none of which we feel must be specifically highlighted here.

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
