# OpenReview forum: "Proteus: Incremental Memory Activation for Long-Context Sequence Modeling"
_ICML.cc/2026/Conference — Submitted to ICML 2026_

### Official Review · Reviewer_Bpyx · 2026-03-13

**Soundness:** 2
**Presentation:** 2
**Significance:** 2
**Originality:** 2
**Overall Recommendation:** 4
**Confidence:** 3

**Summary:**

This paper proposes "Incremental memory activation" for memory-based models, a method to control the effective capacity of memory and progressively expand memory as context grows; and "Proteus", a block-wise gating scheme that applies incremental activation to memory reads and writes. The paper shows improvements when applying Proteus to SOTA models across several benchmarks.

**Compliance With Llm Reviewing Policy:**

Affirmed.

**Final Justification:**

After reading the responses, my main worry about insignificant improvement has been resolved. Additionally, the authors also provided a preliminary theoretical analysis for the method. Thefore, I raise my score to 4.

**Key Questions For Authors:**

1. What is the effect of hyperparameter $E$ in Proteus? This is an important hyperparameter and there should be some experiments in tuning it.
2. Are there any costs when apply Proteus to previous methods? For example, is there additional computing costs?
3. Does Proteus affect the training process in terms of convergence rate and stability when compared to not using Proteus in previous methods?

**Limitations:**

Some limitations of this paper are:
- In terms of results: Not significant improvement over previous methods. This trend is consistent across benchmarks.
- In terms of analysis: Lack of theoretical analysis; Should provide more analysis for hyperparameter $E$ which is important.

**Strengths And Weaknesses:**

Strengths:
- The paper proposes a novel mechanism for memory-based models to improve its long-context memory capabilities.
- The paper is clearly written, well structured and easy to follow for readers.
- The paper includes empirical results on a wide variety of benchmarks.

Weaknesses:
- In most experiments presented, the increase when applying Proteus to previous methods is not significant. For example, across models in Table 1, the increase is only around 0.5.
- With a close gap like that, it is important to perform several runs and report the averaged results and standard deviation, which the paper did not do.
- The paper lacks theoretical analysis for the proposed methods.
- The paper did not provide empirical results when stating that Proteus helps improving continual learning (lines 240-254).

---

> ### Author Rebuttal · Authors · 2026-03-31
>
> We thank the reviewer for the thoughtful feedback. Below we address each concern in turn.
>
> > [W1] The gains are not significant.
>
> We would like to address this concern in two parts.
>
> In Table 1, please note that the absolute value of the improvement is a function of the dataset and benchmark, and that these improvements are considered significant for these benchmark and datasets. For example in similar studies: Miras (ICLR 2026) provides +0.6% improvements over the best recurrent baseline, Mamba-3 (ICLR 2026) show +0.6 improvement over the best baseline,  GatedDeltaNet (ICLR 2025) provides +0.4% gain over its best baseline, and Mamba2 show 0% to 0.5% improvement.
>
> Second, the strongest improvements appear on the long-context evaluations that most directly motivate our method. In Table 2 (Needle-in-a-Haystack), the gains from Proteus become larger as context length increases, which is exactly the regime where static memory saturation and interference are most severe. For example, at 16K context, Proteus improves Titans on S-NIAH-3 from 21.4 to 29.8 and on MK-NIAH-1 from 11.8 to 16.8. We observe the same overall pattern in the additional long-context retrieval results and on LongBench.
>
> > [W2] Average of several runs
>
> We appreciate the comment. However, running multiple seeds for large-scale pretraining is prohibitively expensive, and prior work in this area reports single-run results rather than mean and standard deviation. This is also common in closely related studies such as LaCT, Mamba-3, Miras, Comba, Titans, TTT, Gated DeltaNet, Parallelized DeltaNet, and GSA. In our case, Proteus shows consistent gains over baselines across tasks and datasets, which supports that the improvement is meaningful.
>
> > [W3] Theoretical analysis
>
> We thank the reviewer for the suggestion and have developed a formal theoretical analysis. We summarize the key result here and will include the full version in the revision.
>
> We model the sequence as a Markov chain (a natural model for token dependencies) and the memory as a linear associative memory updated online via gradient descent. Standard online learning theory [Bartlett & Mendelson, 2002; Mohri & Rostamizadeh, 2010] shows that the excess risk (how much worse the memory is compared to optimal) scales proportionally to $c(t)/t$, the ratio of active parameters to tokens seen so far, reflecting the classic bias-variance tradeoff between model capacity and available data [Yu, 1997; Yang & Barron, 1999]. Static memory ($c(t) = C$) therefore has excess risk proportional to $C/t$, which grows large early in the sequence when $t \ll C$, as the memory has far more parameters than tokens seen. A growing schedule ($c(t) \propto t$) keeps this ratio constant, maintaining bounded excess risk throughout.
>
> Under log-loss, which is our training objective (cross-entropy), excess risk directly measures how much predictive information about future tokens the memory loses [Cover & Thomas, 2006], so lower excess risk means the memory retains more of what matters for prediction. The bias-variance optimal schedule from this analysis is $c^*(t) \propto t$, precisely what Proteus implements.
>
> > [W4] Proteus improves continual learning
>
> Our results show that Proteus improves performance across multiple baselines and benchmarks, including LongBench and NIAH, supporting its usefulness for long-context understanding and, therefore, continual learning. That said, this sentence is not a main claim of the paper, and we are happy to remove it if the reviewer finds that it is not well-supported.
>
> > [Q1] The effect of the hyperparameter in Proteus
>
> This is a valuable point, and we thank the reviewer for raising it. We added this analysis by varying the number of partition blocks from 1 (the vanilla model) to 16, and found that performance improved with more blocks.
>
> [Figure - Blocks](https://anonymous.4open.science/r/proteus-figures/blocks.jpg)
>
> > [Q1] Are there any costs in applying Proteus?
>
> Proteus adds no parameters and can be implemented with masking or gating, so only part of the memory is active at each step. Computation is therefore no greater than the baseline, and can be lower.
>
> > [Q2] Convergence rate and stability
>
> Proteus converges at a similar rate to the baseline while reaching a lower final loss, and we do not observe training instability. More broadly, limiting model capacity early in training may help reduce overfitting, since using the full capacity from the start can lead the model to overfit early tokens. Proteus addresses this by limiting early capacity. In our experiments, this did not cause instability. We will clarify this point in the paper.

---

> > ### Author Rebuttal · Reviewer_Bpyx · 2026-04-03
> >
> > I appreciate the authors' hard work and responses.
> >
> > After reading the responses, since the majority of my concerns have been addressed, I will raise my score to 4.

---

> > > ### Author Response · Authors · 2026-04-06
> > >
> > > Thank you for your time and for engaging with our responses. We are pleased that our clarifications addressed your concerns, and we genuinely appreciate the constructive feedback and the updated score.

---

### Official Review · Reviewer_849x · 2026-03-15

**Soundness:** 3
**Presentation:** 3
**Significance:** 3
**Originality:** 4
**Overall Recommendation:** 4
**Confidence:** 4

**Summary:**

1. This paper introduces PROTEUS, a novel paradigm for "incremental memory activation" designed to improve long-context sequence modeling in recurrent neural networks and associative memory systems.
2. The core motivation is that static memory states are often "polluted" by early inputs or saturated by the time later context arrives; PROTEUS addresses this by progressively expanding the effective memory capacity as the sequence length grows.
3. The mechanism employs a lightweight, block-wise gating scheme that restricts memory reads and writes to a small subset of the state initially, unlocking "fresh" capacity over time to reduce interference and encourage better compression.
4. Through extensive testing on architectures like Titans, Hope-Attention, and Comba, the authors demonstrate that PROTEUS consistently improves language modeling perplexity, commonsense reasoning, and long-context retrieval across different model scales.

**Compliance With Llm Reviewing Policy:**

Affirmed.

**Key Questions For Authors:**

1. The experiments are limited to models up to 1.3B parameters and training runs up to 100B tokens, leaving the behavior of PROTEUS on multi-billion parameter "frontier" models unknown.
2. The method assumes that the sequence length is known or that a maximum context length N can be pre-defined for the schedule, which might be a constraint in truly open-ended or streaming applications.

**Limitations:**

see above

**Strengths And Weaknesses:**

Strengths:
1. The paper identifies a fundamental "static memory" limitation in current recurrent models and provides a simple, intuitive, and effective solution that requires no additional parameters.
2. An important aspect assessed by the article is the broad compatibility of PROTEUS, as it is shown to be a "drop-in" improvement for a wide variety of state-of-the-art associative-memory and linear-recurrent architectures.
3. The methodology is highly compute-efficient because it only updates active memory components, potentially saving operations during the early stages of sequence processing.
4. The empirical evaluation is robust, spanning multiple benchmarks including Wikitext, LAMBADA, LongBench, and various Needle-in-a-Haystack configurations, showing gains at both 760M and 1.3B parameter scales.
Weaknesses:
1. The current implementation relies on a deterministic, uniform expansion schedule that may not be optimal for all data types; a data-dependent or learned activation policy is mentioned but not fully explored in this work.
2. The authors proceed to analyze the question of how this applies to pre-training parameters (MLP blocks), but the depth of analysis for this specific "Nested Learning" application is secondary to the memory-specific results.
3. While the paper demonstrates consistent improvements, the performance gains on some reasoning benchmarks are relatively incremental, suggesting the method's impact may vary depending on the specific task requirements.
4. The study primarily focuses on the recurrent/associative memory paradigm, and its relevance to standard sparse or sliding-window attention mechanisms in non-recurrent Transformers is less direct.

---

> ### Author Rebuttal · Authors · 2026-03-31
>
> We thank the reviewer for highlighting the simplicity, efficiency, compatibility, and strong empirical results of Proteus. Below we address the remaining questions.
>
> > [W1] Deterministic, uniform expansion schedule that may not be optimal for all data types
>
> We agree that more sophisticated activation strategies, such as data-dependent or learned schedules, could provide additional benefits. At the same time, one of the main goals of this paper is to introduce the broader paradigm of incremental memory activation in its simplest and cleanest form. Proteus is intended as a proof-of-concept instantiation of this paradigm. We chose a deterministic schedule to isolate the effect of incremental activation itself, without introducing additional complexity from learned controllers or input-dependent policies. As we also discussed in the paper, we agree that richer activation strategies are a promising direction for future work and will clarify this point further in the revision.
>
> > [W2] The depth of analysis for the “Nested Learning” application is secondary to the memory-specific results
>
> The memory-specific setting is the primary focus of the paper, while the pre-training application through the Nested Learning perspective is intended as an additional demonstration that the same incremental activation principle can also be applied beyond online memory updates. In particular, in the pre-training setting, the idea is to schedule which parameter subspace is active for learning, motivated by the Nested Learning view that pre-training itself can be seen as a form of associative compression into model parameters. We will revise the paper to make clearer that this section is a secondary extension of the main memory-centric contribution, rather than an equally developed standalone result.
>
> > [W3] The performance gains on some reasoning benchmarks are relatively incremental
>
> We respectfully emphasize that the gains are significant in this competitive field. Please note that:
>
> (1) Proteus-enhanced models show consistent improvement over their baselines and across multiple datasets, and the absolute value of the improvement is a function of the dataset and benchmark. Please note that these improvements are considered significant for these benchmark and datasets. For example in similar studies: Miras (ICLR 2026) provides +0.6% improvements over the best recurrent baseline, Mamba-3 (ICLR 2026) show +0.6 improvement over the best baseline,  GatedDeltaNet (ICLR 2025) provides +0.4% gain over its best baseline, and Mamba2 show 0% to 0.5% improvement.
>
> (2) Please note that the main goal of language modeling experiments is that we wanted to show that our method does not damage the performance of the model in short context tasks. But the general main motivation which has been improving the performance of the recurrent models in long-context has been supported in other experiments on NIAH, long-context retrieval, and longbench, showing clear improvement.
>
> > [W4] Relevance to standard sparse or sliding-window attention mechanisms in non-recurrent Transformers is less direct
>
> Please note that sparse attention and other *softmax* attention variants are orthogonal to this direction. Our method targets recurrent architectures and their memory that is based on compression. sparse or dense *softmax* attention variants are based on caching tokens and so there is no compression involved. Please let us know if we can provide more information or clarification.
>
> > [Q1] The experiments are limited to models up to 1.3B parameters and training runs up to 100B tokens
>
> We agree that evaluation at frontier scale would be valuable. In this work, however, Proteus is studied as a new pre-training architectural design, and all four baselines and their corresponding Proteus variants were trained fully from scratch at two substantial scales: 760M parameters / 50B tokens and 1.3B parameters / 100B tokens. Extending this study to multi-billion-parameter frontier models is beyond our current computational budget, but we view it as an important direction for future work.
>
> > [Q2] The method assumes that the sequence length is known or that a maximum context length N can be pre-defined for the schedule
>
> That is a great question. Please note that even in the current design of memory models, all parameters are active from the begining and Proteus can at least delays this process and improve upon existing designs. We acknowledge the importance of future studies to further improve the design of incremental parameter activation, mainly by designing more sophisticated scheduling that are invariant with respect to the maximum context length.

---

> > ### Author Rebuttal · Reviewer_849x · 2026-04-07
> >
> > My concerns have been addressed. I would like to keep the current score.

---

> > > ### Author Response · Authors · 2026-04-07
> > >
> > > Thank you for your thorough review and for engaging with our rebuttal. Your feedback was constructive and will help us improve the paper.

---

### Official Review · Reviewer_x8Yg · 2026-03-18

**Soundness:** 2
**Presentation:** 2
**Significance:** 2
**Originality:** 2
**Overall Recommendation:** 4
**Confidence:** 2

**Summary:**

This paper presents PROTEUS, an incremental memory activation mechanism that addresses static memory saturation in long-context models. By progressively expanding capacity via block-wise gating, it enhances historical compression while reducing interference. Validated across four distinct architectures, PROTEUS provides consistent performance gains with zero computational overhead, offering a highly practical advancement for recurrent sequence modeling.

**Compliance With Llm Reviewing Policy:**

Affirmed.

**Final Justification:**

The author addressed most of my concerns.

**Key Questions For Authors:**

See weakness

**Strengths And Weaknesses:**

**Strengths**

1. The method improves length extrapolation and retrieval accuracy in long-context tasks, such as NIAH, with performance gains becoming more pronounced as sequence length increases.

2. By utilizing incremental activation, PROTEUS prevents "memory pollution" from early inputs and reduces interference by progressively unlocking fresh capacity as the context grows.

**Weakness**

1. The current activation schedule is deterministic, lacking data-dependent policies that could dynamically adapt the memory capacity based on the specific importance of input content.

2. The evaluation lacks comprehensive sensitivity analysis for the partition parameter $E$, which was fixed at four blocks for all experiments, leaving the optimal block count unexplored .

3. The author claimed that their method is a lightweight, drop-in method. But they only provided experiments on 1.3B size, which cannot convince me about the scalability of their method.

4. The experimental results seem weird. Many models reach the ppl of over 20, which means that they hardly have the ability of language modeling, while they still can achieve well performance on down-stream tasks.

---

> ### Author Rebuttal · Authors · 2026-03-31
>
> Thank you for your time and detailed feedback. We answer your questions below:
>
>
> > [W1] Data-dependent policies for memory activation
>
> Please note that we have explicitly mentioned this in the initial submission as one of the future directions, and noted in the paper that the current schedule is a simple proof of concept for a more general incremental activation framework, and that richer input-dependent policies are a natural next step. Our goal in this paper is twofold: first, to introduce the general principle of incremental memory activation; and second, to provide a clean proof-of-concept instantiation in Proteus, and support its effectiveness. We chose a deterministic schedule to isolate the effect of the paradigm itself without introducing additional complexity from learned or content-dependent controllers.
>
>
> > [W2] Comprehensive sensitivity analysis for the partition parameter
>
> This is an important point, and we appreciate the reviewer for raising it. In response, we have added an ablation that varies the number of partition blocks from 1 (which recovers the vanilla model) up to 16. We found that performance improved with more blocks, and we will include these results in the revision and discuss them.
>
> [Figure - Blocks](https://anonymous.4open.science/r/proteus-figures/blocks.jpg)
>
> > [W3] The authors claim that the method is lightweight and drop-in, but only provide experiments at 1.3B scale
>
> We thank the reviewer for raising this point. By “lightweight, drop-in,” we mean that Proteus can be incorporated into an existing architecture design with minimal modification when training from scratch. We do not mean that it can be added “post hoc” to an already trained model. In our setting, Proteus is implemented through simple masking/gating over existing memory components, without introducing a separate heavy module or changing the backbone architecture. As a result, a wide range of architecture designs can be adapted to follow the incremental memory activation paradigm.
>
> We would also like to clarify that our paper reports results at two scales: 760M parameters / 50B tokens and 1.3B parameters / 100B tokens. Since all baseline models and their corresponding Proteus variants are trained from scratch, these experiments already require substantial computational resources. We agree that validation at even larger scales would further strengthen the scalability claim, but such runs are beyond our current compute budget.
>
> > [W4] Perplexity of over 20
>
> Please note that the results presented in the paper for the baselines  are pretty standard in the literature and in fact the Proteus-enhanced variants are providing meaningful improvement over the baselines. In general, we have followed existing similar works and common practice in the literature, and trained models in academic scale and on 100B tokens. Definately, using more tokens and larger models can enahnce the perplexity. Also, please note that still these models have language modeling capapbilities and have been used with similar sizes and setups in similar papers such as: LaCT (ICLR 2026), Miras (ICLR 2026), Comba (ICLR 2026), Titans (NeurIPS 2025),  TTT (ICML 2025), Gated DeltaNet (ICLR 2025), Parallelized DeltaNet (NeurIPS 2025), and many more.

---

> > ### Author Rebuttal · Reviewer_x8Yg · 2026-04-02
> >
> > The author addressed most of my concerns, except that they left one for future works. After reading them, I will raise my score to 4.

---

> > > ### Author Response · Authors · 2026-04-06
> > >
> > > Thank you for carefully reading our rebuttal and for raising your score. We are glad the clarifications and additional experiments were helpful, and we truly appreciate your time and engagement throughout the review process.

---

### Official Review · Reviewer_NhdT · 2026-03-18

**Soundness:** 3
**Presentation:** 3
**Significance:** 3
**Originality:** 3
**Overall Recommendation:** 5
**Confidence:** 4

**Summary:**

The paper tackles the static memory capacity in RNNs and associative memory models. In traditional models, the entire hidden state is available from the first token, resulting in earlier tokens stored with little compression, while later tokens must be highly compressed or overwrite existing information. The authors propose "incremental memory activation", a paradigm where memory capacity grows as a function of context length. This is instantiated via PROTEUS, a block-wise gating mechanism that can be easily implemented in existing frameworks and architectures. PROTEUS 1) imposes a bottleneck on early inputs by reducing the available memory state and 2) progressively unlocks unused memory for later tokens, reducing interference between new and old inputs. The method shows consistent improvements in language modeling, common-sense reasoning, and long-context tasks.

**Compliance With Llm Reviewing Policy:**

Affirmed.

**Final Justification:**

The authors addressed my concerns and curiosities.

**Key Questions For Authors:**

Other than the doubts mentioned above, I ask the authors to address the following:

* The choice of the activation schedule is very arbitrary. While the author explained that they leave the study of other strategies (non-deterministic, input dependent, etc) to future works, it would be better to include some ablations on the impact of the number of partition blocks $E$ on overall performances.

* How does this approach, especially at the beginning of training, differ from models that grow in depth? Specifically, if the goal is to avoid using memory until it is needed, and since the section of the memory used is uniformly decided, why not simply remove those parameters in the first part of the sequence?

* The authors claim that PROTEUS helps with continual learning by updating only a small number of parameters and mitigating catastrophic forgetting. How does this compare empirically to established methods like LoRA?

* Could the author provide plots of the training loss or dynamics for the first tokens? I am wondering if the model is simply overfitting faster on these inputs, given that it has fewer active parameters to manage initially.

**Limitations:**

yes

**Strengths And Weaknesses:**

# Strengths

* The proposed framework PROTEUS can be easily incorporated into a vast class of standard and model architectures.

* The authors provide a wide range of experiments across scales and tests on various capabilities, showing that PROTEUS consistently outperforms base models on benchmarks. In particular, it improves length extrapolation and retrieval in long sequences, where traditionally RNNs struggle more.

# Weaknesses

* The paper applies PROTEUS to the MLP block during pre-training. However, existing literature [1] suggests MLPs handle general knowledge while the sequence mixer handles context. The intuition for why the incremental activation is optimal for the "general knowledge" component could be investigated in more depth, with some either theoretical or experimental results.

* Natural language often prioritises recent context following a power law (as suggested in [2]). By forcing heavy compression on the earliest tokens, is there a risk of going against this bias? The paper would benefit from showing the specific trade-offs in recall performances for early vs late tokens when using PROTEUS (such as showing how perplexity changes based on the position of the tokens in the sequence).

* The performance gains could be an artifact of PROTEUS creating a smaller (and so easier to optimize) model during the initial training phase. The paper doesn't provide enough insights into how these dynamics change compared to baselines or similar "growing" models as in [3], especially on earlier tokens.

[1] Understanding Transformer from the Perspective of Associative Memory (https://arxiv.org/abs/2505.19488v1)

[2] Zoology: Measuring and Improving Recall in Efficient Language Models (https://arxiv.org/abs/2312.04927)

[3] From Growing to Looping: A Unified View of Iterative Computation in LLMs (https://arxiv.org/abs/2602.16490)

---

> ### Author Rebuttal · Authors · 2026-03-31
>
> Thank you for your time and detailed feedback. We answer your questions below:
>
> > [W1] Incremental activation appropriate and “general knowledge”
>
> We want to kindly bring two points to your attention: (1) A major part of our contribution is to use Proteus for the associative memory modules such as recurrent neural networks. Later, we also provide motivations and emprical evidences that this technique can be applied for **adaptive** MLP blocks in designs such as Hope-attention. Please note that MLP blocks in Hope-attention are beyound MLP blocks in Transformers, and so "general knowledge". In fact, they store general knowledge while they also change in-context. We use Proteus for the second part, when MLP blocks are also updated based on the context. (2) Even considering storing "general knowledge" in MLP blocks, it is impossible to store all knowledge of web data into the parameters of an MLP and so there is a compression process involved. In this compression process, earlier data samples have more capacity to be compressed in, while later data samples have less capacity. Using Proteus, the general knowldge stored in earlier and later data samples can have relatively equal free capacity for compression.
>
> > [W2] Recency bias and Perplexity by token position
>
> Proteus compresses early tokens more aggressively and reserves capacity for later ones. This does not oppose recency bias; it makes the tradeoff explicit. The goal is not to remove recency effects, but to avoid static-capacity memories overallocating degrees of freedom to early inputs and causing interference for later information. Our results suggest this tradeoff is beneficial overall, especially in long-context retrieval settings (Tables 2–3, Figure 2), where later information is hardest to preserve.
>
> We agree that position-wise analysis would strengthen the paper. We therefore added a perplexity-by-token-index comparison between the baseline and Proteus variants. The new results show that Proteus achieves lower perplexity across all token positions.
>
> [Figure - Perplexity](https://anonymous.4open.science/r/proteus-figures/ppl.jpg)
>
> > [W3] Maybe it is just a smaller/easier model early
>
> Proteus reduces the active subspace early in the sequence to achieve better compression, but it is not equivalent to training a smaller model. Inactive blocks are not removed; they are preserved and progressively unlocked, so reads and writes are restricted only temporarily. Proteus is therefore a position-dependent capacity-allocation policy, not a reduction in final model size. In pretraining, this corresponds to scheduling which parameters are updated rather than deleting parameters.
>
> We show that Proteus improves performance, and our ablations attribute these gains to incremental activation. The reviewer’s cited baseline is from February 2026, after the ICML submission deadline.
>
> > [Q1] Impact of the number of partition blocks
>
> We agree this is an important missing ablation. We evaluated Proteus with (E) ranging from 1 (the vanilla model) to 16, and found performance improved with more blocks. We will include this ablation in the revision.
>
> [Figure - Blocks](https://anonymous.4open.science/r/proteus-figures/blocks.jpg)
>
> > [Q2] If the goal is to avoid using memory until it is needed...
>
> We would like to clarify that the goal is not to defer memory use, but to prevent early tokens from consuming too much capacity when memory is still empty. Proteus restricts early reads and writes to encourage stronger compression, then progressively unlocks more capacity while preserving stored information. This lets later tokens benefit from earlier context while still retaining fresh capacity for writing.
>
> > [Q3] Continual learning vs LoRA
>
> Great question. Please note that methods like Proteus and LoRA are effectively orthogonal, meaning they can be applied together without interfering with each other. Accordingly, they are not directly comparable approaches. In fact, to avoid catastrophic forgetting, one might consider two strategies: (1) increasing the capacity for new data samples to reduce interference between new and old data, and/or (2) updating fewer parameters to reduce forgetting. Proteus addresses (1) by incrementally activating parameters and providing larger capacity for later tokens or data samples. In contrast, methods like LoRA target (2): they do not affect model capacity, but instead update only a small number of parameters so that the model forgets less. In principle, one could combine both methods to address both (1) and (2).
> > [Q4] First-token dynamics / overfitting
>
> More broadly, using fewer parameters for early data helps prevent overfitting: if full capacity is available from the start, the model may overfit to the initial tokens. Proteus mitigates this by restricting early capacity. In our experiments, we did not observe instability, and regarding token-position dynamics, we showed consistently lower perplexity across token positions in response to W2.

---

> > ### Author Rebuttal · Reviewer_NhdT · 2026-04-03
> >
> > I thank the reviewer for the clarification, the new experiments and for improving their work even more. All my doubts are cleared and I am happy to increase my score to 5

---

> > > ### Author Response · Authors · 2026-04-06
> > >
> > > Thank you for your thorough engagement throughout the review process and for taking the time to carefully evaluate our responses and the additional experiments. We are glad the clarifications and new results addressed your concerns, and we appreciate your constructive feedback.

---

### Decision · Program_Chairs · 2026-04-30

**Decision:**

Reject

**Comment:**

The paper addresses a core problem in language modeling: increasing memory capacity in sequence models. This is done by scheduling memory and adding a gating mechanism to control the write mechanisms. In this way, the model incrementally expands the active subset of the memory state as the context progresses. The proposed approach can be applied to several architectures and is, in principle, a valuable contribution. Yet, while the method is promising, my biggest concern is that the paper crucially misses the empirical comparison with "Memory caching: RNNs with growing memory, 2026." -- cited by the authors. This reference is motivated by similar principles and discusses performance boosts in setups akin to those explored by the authors here. A comparison with log-linear attention (cited by the authors) is also missing and is important, as this approach is closely related in spirit yet less general. I suggest the authors take the literature into full consideration for their comparisons and resubmit.